# FAST IS BETTER THAN FREE:
# REVISITING ADVERSARIAL TRAINING

**Eric Wong**[*]
Machine Learning Department
Carnegie Mellon University
Pittsburgh, PA 15213, USA
ericwong@cs.cmu.edu

**Leslie Rice**[*]
Computer Science Department
Carnegie Mellon University
Pittsburgh, PA 15213, USA
larice@cs.cmu.edu

**J. Zico Kolter**
Computer Science Department
Carnegie Mellon University and
Bosch Center for Artifical Intelligence
Pittsburgh, PA 15213, USA
zkolter@cs.cmu.edu

## ABSTRACT

Adversarial training, a method for learning robust deep networks, is typically assumed to be more expensive than traditional training due to the necessity of constructing adversarial examples via a first-order method like projected gradient decent (PGD). In this paper, we make the surprising discovery that it is possible to train empirically robust models using a much weaker and cheaper adversary, an approach that was previously believed to be ineffective, rendering the method no more costly than standard training in practice. Specifically, we show that adversarial training with the fast gradient sign method (FGSM), when combined with random initialization, is as effective as PGD-based training but has significantly lower cost. Furthermore we show that FGSM adversarial training can be further accelerated by using standard techniques for efficient training of deep networks, allowing us to learn a robust CIFAR10 classifier with 45% robust accuracy to PGD attacks with $\epsilon = 8/255$ in 6 minutes, and a robust ImageNet classifier with 43% robust accuracy at $\epsilon = 2/255$ in 12 hours, in comparison to past work based on "free" adversarial training which took 10 and 50 hours to reach the same respective thresholds. Finally, we identify a failure mode referred to as "catastrophic overfitting" which may have caused previous attempts to use FGSM adversarial training to fail. All code for reproducing the experiments in this paper as well as pretrained model weights are at https://github.com/locuslab/fast_adversarial.

## 1 INTRODUCTION

Although deep network architectures continue to be successful in a wide range of applications, the problem of learning *robust* deep networks remains an active area of research. In particular, safety and security focused applications are concerned about robustness to adversarial examples, data points which have been adversarially perturbed to fool a model (Szegedy et al., 2013). The goal here is to learn a model which is not only accurate on the data, but also accurate on adversarially perturbed versions of the data. To this end, a number of defenses have been proposed to mitigate the problem and improve the robustness of deep networks, with some of the most reliable being certified defenses and adversarial training. However, both of these approaches come at a non-trivial, additional computational cost, often increasing training time by an order of magnitude over standard training. This has slowed progress in researching robustness in deep networks, due to the computational difficulty in scaling to much larger networks and the inability to rapidly train models when experimenting with new ideas. In response to this difficulty, there has been a recent surge in work

---

[*]Equal contribution.

that tries to to reduce the complexity of generating an adversarial example, which forms the bulk of the additional computation in adversarial training (Zhang et al., 2019; Shafahi et al., 2019). While these works present reasonable improvements to the runtime of adversarial training, they are still significantly slower than standard training, which has been greatly accelerated due to competitions for optimizing both the speed and cost of training (Coleman et al., 2017).

In this work, we argue that adversarial training, in fact, is not as hard as has been suggested by this past line of work. In particular, we revisit one of the the *first* proposed methods for adversarial training, using the Fast Gradient Sign Method (FGSM) to add adversarial examples to the training process (Goodfellow et al., 2014). Although this approach has long been dismissed as ineffective, we show that by simply introducing random initialization points, FGSM-based training is *as effective as projected gradient descent based training* while being an order of magnitude more efficient. Moreover, FGSM adversarial training (and to a lesser extent, other adversarial training methods) can be drastically accelerated using standard techniques for efficient training of deep networks, including e.g. cyclic learning rates (Smith & Topin, 2018), mixed-precision training (Micikevicius et al., 2017), and other similar techniques. The method has extremely few free parameters to tune, and can be easily adapted to most training procedures. We further identify a failure mode that we call "catastrophic overfitting", which may have caused previous attempts at FGSM adversarial training to fail against PGD-based attacks.

The end result is that, with these approaches, we are able to train (empirically) robust classifiers far faster than in previous work. Specifically, we train an $\ell_\infty$ robust CIFAR10 model to $45\%$ accuracy at $\epsilon = 8/255$ (the same level attained in previous work) in *6 minutes*; previous papers reported times of 80 hours for PGD-based training (Madry et al., 2017) and 10 hours for the more recent "free" adversarial training method (Shafahi et al., 2019). Similarly, we train an $\ell_\infty$ robust ImageNet classifier to $43\%$ top-1 accuracy at $\epsilon = 2/255$ (again matching previous results) in 12 hours of training (compared to 50 hours in the best reported previous work that we are aware of (Shafahi et al., 2019)). Both of these times roughly match the comparable time for quickly training a standard *non-robust* model to reasonable accuracy. We extensively evaluate these results against *strong PGD-based attacks*, and show that they obtain the same empirical performance as the slower, PGD-based training. Thus, we argue that despite the conventional wisdom, adversarially robust training is not actually more challenging than standard training of deep networks, and can be accomplished with the notoriously weak FGSM attack.

## 2    RELATED WORK

After the discovery of adversarial examples by Szegedy et al. (2013), Goodfellow et al. (2014) proposed the Fast Gradient Sign Method (FGSM) to generate adversarial examples with a single gradient step. This method was used to perturb the inputs to the model before performing back-propagation as an early form of adversarial training. This attack was enhanced by adding a randomization step, which was referred to as R+FGSM (Tramèr et al., 2017). Later, the Basic Iterative Method improved upon FGSM by taking multiple, smaller FGSM steps, ultimately rendering both FGSM-based adversarial training ineffective (Kurakin et al., 2016). This iterative adversarial attack was further strengthened by adding multiple random restarts, and was also incorporated into the adversarial training procedure. These improvements form the basis of what is widely understood today as adversarial training against a projected gradient descent (PGD) adversary, and the resulting method is recognized as an effective approach to learning robust networks (Madry et al., 2017). Since then, the PGD attack and its corresponding adversarial training defense have been augmented with various techniques, such as optimization tricks like momentum to improve the adversary (Dong et al., 2018), combination with other heuristic defenses like matrix estimation (Yang et al., 2019) or logit pairing (Mosbach et al., 2018), and generalization to multiple types of adversarial attacks (Tramèr & Boneh, 2019; Maini et al., 2019).

In addition to adversarial training, a number of other defenses against adversarial attacks have also been proposed. Adversarial defenses span a wide range of methods, such as preprocessing techniques (Guo et al., 2017; Buckman et al., 2018; Song et al., 2017), detection algorithms (Metzen et al., 2017; Feinman et al., 2017; Carlini & Wagner, 2017a), verification and provable defenses (Katz et al., 2017; Sinha et al., 2017; Wong & Kolter, 2017; Raghunathan et al., 2018), and various theoretically motivated heuristics (Xiao et al., 2018; Croce et al., 2018). While certified defenses

have been scaled to reasonably sized networks (Wong et al., 2018; Mirman et al., 2018; Gowal et al., 2018; Cohen et al., 2019; Salman et al., 2019), the guarantees don't match the empirical robustness obtained through adversarial training.

With the proposal of many new defense mechanisms, of great concern in the community is the use of strong attacks for evaluating robustness: weak attacks can give a misleading sense of security, and the history of adversarial examples is littered with adversarial defenses (Papernot et al., 2016; Lu et al., 2017; Kannan et al., 2018; Tao et al., 2018) which were ultimately defeated by stronger attacks (Carlini & Wagner, 2016; 2017b; Athalye et al., 2017; Engstrom et al., 2018; Carlini, 2019). This highlights the difficulty of evaluating adversarial robustness, as pointed out by other work which began to defeat proposed defenses en masse (Uesato et al., 2018; Athalye et al., 2018). Since then, several best practices have been proposed to mitigate this problem (Carlini et al., 2019).

Despite the eventual defeat of other adversarial defenses, adversarial training with a PGD adversary remains empirically robust to this day. However, running a strong PGD adversary within an inner loop of training is expensive, and some earlier work in this topic found that taking larger but fewer steps did not always significantly change the resulting robustness of a network (Wang, 2018). To combat the increased computational overhead of the PGD defense, some recent work has looked at regressing the $k$-step PGD adversary to a variation of its single-step FGSM predecessor called "free" adversarial training, which can be computed with little overhead over standard training by using a single backwards pass to simultaneously update both the model weights and also the input perturbation (Shafahi et al., 2019). Finally, when performing a multi-step PGD adversary, it is possible to cut out redundant calculations during backpropagation when computing adversarial examples for additional speedup (Zhang et al., 2019).

Although these improvements are certainly faster than the standard adversarial training procedure, they are not much faster than traditional training methods, and can still take hours to days to compute. On the other hand, top performing training methods from the DAWNBench competition (Coleman et al., 2017) are able to train CIFAR10 and ImageNet architectures to standard benchmark metrics in mere minutes and hours respectively, using only a modest amount of computational resources. Although some of the techniques can be quite problem specific for achieving bleeding-edge performance, more general techniques such as cyclic learning rates (Smith & Topin, 2018) and half-precision computations (Micikevicius et al., 2017) have been quite successful in the top ranking submissions, and can also be useful for adversarial training.

## 3 ADVERSARIAL TRAINING OVERVIEW

Adversarial training is a method for learning networks which are robust to adversarial attacks. Given a network $f_\theta$ parameterized by $\theta$, a dataset $(x_i, y_i)$, a loss function $\ell$ and a threat model $\Delta$, the learning problem is typically cast as the following robust optimization problem,

$$\min_\theta \sum_i \max_{\delta \in \Delta} \ell(f_\theta(x_i + \delta), y_i). \tag{1}$$

A typical choice for a threat model is to take $\Delta = \{\delta : \|\delta\|_\infty \leq \epsilon\}$ for some $\epsilon > 0$. This is the $\ell_\infty$ threat model used by Madry et al. (2017) and is the setting we study in this paper. The procedure for adversarial training is to use some adversarial attack to approximate the inner maximization over $\Delta$, followed by some variation of gradient descent on the model parameters $\theta$. For example, one of the earliest versions of adversarial training used the Fast Gradient Sign Method to approximate the inner maximization. This could be seen as a relatively inaccurate approximation of the inner maximization for $\ell_\infty$ perturbations, and has the following closed form (Goodfellow et al., 2014):

$$\delta^\star = \epsilon \cdot \text{sign}(\nabla_x \ell(f(x), y)). \tag{2}$$

A better approximation of the inner maximization is to take multiple, smaller FGSM steps of size $\alpha$ instead. When the iterate leaves the threat model, it is projected back to the set $\Delta$ (for $\ell_\infty$ perturbations. This is equivalent to clipping $\delta$ to the interval $[-\epsilon, \epsilon]$). Since this is only a local approximation of a non-convex function, multiple random restarts within the threat model $\Delta$ typically improve the approximation of the inner maximization even further. A combination of all these techniques is

---

**Algorithm 1** PGD adversarial training for $T$ epochs, given some radius $\epsilon$, adversarial step size $\alpha$ and $N$ PGD steps and a dataset of size $M$ for a network $f_\theta$

---

**for** $t = 1 \ldots T$ **do**
    **for** $i = 1 \ldots M$ **do**
        *// Perform PGD adversarial attack*
        $\delta = 0$ *// or randomly initialized*
        **for** $j = 1 \ldots N$ **do**
            $\delta = \delta + \alpha \cdot \text{sign}(\nabla_\delta \ell(f_\theta(x_i + \delta), y_i))$
            $\delta = \max(\min(\delta, \epsilon), -\epsilon)$
        **end for**
        $\theta = \theta - \nabla_\theta \ell(f_\theta(x_i + \delta), y_i)$ *// Update model weights with some optimizer, e.g. SGD*
    **end for**
**end for**

---

**Algorithm 2** "Free" adversarial training for $T$ epochs, given some radius $\epsilon$, $N$ minibatch replays, and a dataset of size $M$ for a network $f_\theta$

---

$\delta = 0$
*// Iterate T/N times to account for minibatch replays and run for T total epochs*
**for** $t = 1 \ldots T/N$ **do**
    **for** $i = 1 \ldots M$ **do**
        *// Perform simultaneous FGSM adversarial attack and model weight updates T times*
        **for** $j = 1 \ldots N$ **do**
            *// Compute gradients for perturbation and model weights simultaneously*
            $\nabla_\delta, \nabla_\theta = \nabla \ell(f_\theta(x_i + \delta), y_i)$
            $\delta = \delta + \epsilon \cdot \text{sign}(\nabla_\delta)$
            $\delta = \max(\min(\delta, \epsilon), -\epsilon)$
            $\theta = \theta - \nabla_\theta$ *// Update model weights with some optimizer, e.g. SGD*
        **end for**
    **end for**
**end for**

---

known as the PGD adversary (Madry et al., 2017), and its usage in adversarial training is summarized in Algorithm 1.

Note that the number of gradient computations here is proportional to $O(MN)$ in a single epoch, where $M$ is the size of the dataset and $N$ is the number of steps taken by the PGD adversary. This is $N$ times greater than standard training (which has $O(M)$ gradient computations per epoch), and so adversarial training is typically $N$ times slower than standard training.

### 3.1 "FREE" ADVERSARIAL TRAINING

To get around this slowdown of a factor of $N$, Shafahi et al. (2019) instead propose "free" adversarial training. This method takes FGSM steps with full step sizes $\alpha = \epsilon$ followed by updating the model weights for $N$ iterations on the same minibatch (also referred to as "minibatch replays"). The algorithm is summarized in Algorithm 2. Note that perturbations are not reset between minibatches. To account for the additional computational cost of minibatch replay, the total number of epochs is reduced by a factor of $N$ to make the total cost equivalent to $T$ epochs of standard training. Although "free" adversarial training is faster than the standard PGD adversarial training, it is not as fast as we'd like: Shafahi et al. (2019) need to run over 200 epochs in over 10 hours to learn a robust CIFAR10 classifier and two days to learn a robust ImageNet classifier, whereas standard training can be accomplished in minutes and hours for the same respective tasks.

## 4 FAST ADVERSARIAL TRAINING

To speed up adversarial training and move towards the state of the art in fast standard training methods, we first highlight the main empirical contribution of the paper: that FGSM adversarial

---

**Algorithm 3** FGSM adversarial training for $T$ epochs, given some radius $\epsilon$, $N$ PGD steps, step size $\alpha$, and a dataset of size $M$ for a network $f_\theta$

---

  **for** $t = 1 \ldots T$ **do**
    **for** $i = 1 \ldots M$ **do**
      *// Perform FGSM adversarial attack*
      $\delta = \text{Uniform}(-\epsilon, \epsilon)$
      $\delta = \delta + \alpha \cdot \text{sign}(\nabla_\delta \ell(f_\theta(x_i + \delta), y_i))$
      $\delta = \max(\min(\delta, \epsilon), -\epsilon)$
      $\theta = \theta - \nabla_\theta \ell(f_\theta(x_i + \delta), y_i)$ *// Update model weights with some optimizer, e.g. SGD*
    **end for**
  **end for**

---

Table 1: Standard and robust performance of various adversarial training methods on CIFAR10 for $\epsilon = 8/255$ and their corresponding training times

| Method | Standard accuracy | PGD ($\epsilon = 8/255$) | Time (min) |
|---|---|---|---|
| FGSM + DAWNBench | | | |
|   + zero init | 85.18% | 0.00% | 12.37 |
|     + early stopping | 71.14% | 38.86% | 7.89 |
|   + previous init | 86.02% | 42.37% | 12.21 |
|   + random init | 85.32% | 44.01% | 12.33 |
|     + $\alpha = 10/255$ step size | 83.81% | 46.06% | 12.17 |
|     + $\alpha = 16/255$ step size | 86.05% | 0.00% | 12.06 |
|       + early stopping | 70.93% | 40.38% | 8.81 |
| "Free" ($m = 8$) (Shafahi et al., 2019)[1] | 85.96% | 46.33% | 785 |
|   + DAWNBench | 78.38% | 46.18% | 20.91 |
| PGD-7 (Madry et al., 2017)[2] | 87.30% | 45.80% | 4965.71 |
|   + DAWNBench | 82.46% | 50.69% | 68.8 |

training combined with random initialization is just as effective a defense as PGD-based training. Following this, we discuss several techniques from the DAWNBench competition (Coleman et al., 2017) that are applicable to all adversarial training methods, which reduce the total number of epochs needed for convergence with cyclic learning rates and further speed up computations with mixed-precision arithmetic.

## 4.1 REVISITING FGSM ADVERSARIAL TRAINING

Despite being quite similar to FGSM adversarial training, free adversarial training is empirically robust against PGD attacks whereas FGSM adversarial training is not believed to be robust. To analyze why, we identify a key difference between the methods: a property of free adversarial training is that the perturbation from the previous iteration is used as the initial starting point for the next iteration. However, there is little reason to believe that an adversarial perturbation for a previous minibatch is a reasonable starting point for the next minibatch. As a result, we hypothesize that the main benefit comes from simply starting from a non-zero initial perturbation.

In light of this difference, our approach is to use FGSM adversarial training with random initialization for the perturbation, as shown in Algorithm 3. We find that, in contrast to what was previously believed, this simple adjustment to FGSM adversarial training can be used as an effective defense on par with PGD adversarial training. Crucially, we find that starting from a non-zero initial perturbation is the primary driver for success, regardless of the actual initialization. In fact, both starting with the previous minibatch's perturbation or initializing from a uniformly random perturbation al-

---

[1] As reported by Shafahi et al. (2019) using a different network architecture and an adversary with 20 steps and 10 restarts, which is strictly weaker than the adversary used in this paper.

[2] As reported by Madry et al. (2017) using a different network architecture and an adversary and an adversary with 20 steps and no restarts, which is strictly weaker than the adversary used in this paper

low FGSM adversarial training to succeed at being robust to full-strength PGD adversarial attacks. Note that randomized initialization for FGSM is not a new idea and was previously studied by Tramèr et al. (2017). Crucially, Tramèr et al. (2017) use a different, more restricted random initialization and step size, which does not result in models robust to full-strength PGD adversaries. A more detailed comparison of their approach with ours is in Appendix A.

To test the effect of initialization in FGSM adversarial training, we train several models to be robust at a radius $\epsilon = 8/255$ on CIFAR10, starting with the most "pure" form of FGSM, which takes steps of size $\alpha = \epsilon$ from a zero-initialized perturbation. The results, given in Table 1, are consistent with the literature, and show that the model trained with zero-initialization is not robust against a PGD adversary. However, surprisingly, simply using a random or previous-minibatch initialization instead of a zero initialization actually results in reasonable robustness levels (with random initialization performing slightly better) that are comparable to both free and PGD adversarial training methods. The adversarial accuracies in Table 1 are calculated using a PGD adversary with 50 iterations, step size $\alpha = 2/255$, and 10 random restarts. Specific optimization parameters used for training these models can be found in Appendix B.

**FGSM step size**   Note that an FGSM step with size $\alpha = \epsilon$ from a non-zero initialization is not guaranteed to lie on the boundary of the $\ell_\infty$ ball, and so this defense could potentially be seen as too weak. We find that increasing the step size by a factor of 1.25 to $\alpha = 10/255$ further improved the robustness of the model so that it is on par with the best reported result from free adversarial training. However, we also found that forcing the resulting perturbation to lie on the boundary with a step size of $\alpha = 2\epsilon$ resulted in catastrophic overfitting: it does not produce a model robust to adversarial attacks. These two failure modes (starting from a zero-initialized perturbation and generating perturbations at the boundary) may explain why previous attempts at FGSM adversarial training failed, as the model overfits to a restricted threat model, and is described in more detail in Section 5.4. A full curve showing the effect of a range of FGSM step sizes on the robust performance can be found in Appendix C.

**Computational complexity**   A second key difference between FGSM and free adversarial training is that the latter uses a single backwards pass to compute gradients for both the perturbation and the model weights while repeating the same minibatch $m$ times in a row, called "minibatch replay". In comparison, the FGSM adversarial training does not need to repeat minibatches, but needs two backwards passes to compute gradients separately for the perturbation and the model weights. As a result, the computational complexity for an epoch of FGSM adversarial training is not truly free and is equivalent to two epochs of standard training.

## 4.2   DAWNBench improvements

Although free adversarial training is of comparable cost per iteration to traditional standard training methods, it is not quite comparable in total cost to more recent advancements in fast methods for standard training. Notably, top submissions to the DAWNBench competition have shown that CIFAR10 and ImageNet classifiers can be trained at significantly quicker times and at much lower cost than traditional training methods. Although some of the submissions can be quite unique in their approaches, we identify two generally applicable techniques which have a significant impact on the convergence rate and computational speed of standard training.

**Cyclic learning rate**   Introduced by Smith (2017) for improving convergence and reducing the amount of tuning required when training networks, a cyclic schedule for a learning rate can drastically reduce the number of epochs required for training deep networks (Smith & Topin, 2018). A simple cyclic learning rate schedules the learning rate linearly from zero, to a maximum learning rate, and back down to zero (examples can be found in Figure 1). Using a cyclic learning rate allows CIFAR10 architectures to converge to benchmark accuracies in tens of epochs instead of hundreds, and is a crucial component of some of the top DAWNBench submissions.

**Mixed-precision arithmetic**   With newer GPU architectures coming with tensor cores specifically built for rapid half-precision calculations, using mixed-precision arithmetic when training deep networks can also provide significant speedups for standard training (Micikevicius et al., 2017). This

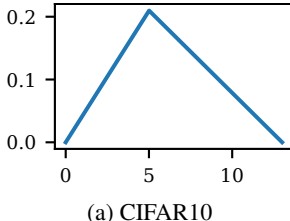

(a) CIFAR10

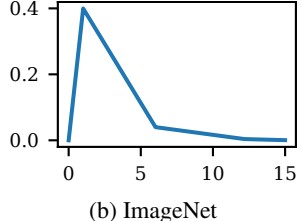

(b) ImageNet

Figure 1: Cyclic learning rates used for FGSM adversarial training on CIFAR10 and ImageNet over epochs. The ImageNet cyclic schedule is decayed further by a factor of 10 in the second and third phases.

Table 2: Robustness of FGSM and PGD adversarial training on MNIST

| Method | Standard accuracy | PGD ($\epsilon = 0.1$) | PGD ($\epsilon = 0.3$) | Verified ($\epsilon = 0.1$) |
|---|---|---|---|---|
| PGD | 99.20% | 97.66% | 89.90% | 96.7% |
| FGSM | 99.20% | 97.53% | 88.77% | 96.8% |

can drastically reduce the memory utilization, and when tensor cores are available, also reduce runtime. In some DAWNBench submissions, switching to mixed-precision computations was key to achieving fast training while keeping costs low.

We adopt these two techniques for use in adversarial training, which allows us to drastically reduce the number of training epochs as well as the runtime on GPU infrastructure with tensor cores, while using modest amounts of computational resources. Notably, both of these improvements can be easily applied to existing implementations of adversarial training by adding a few lines of code with very little additional engineering effort, and so are easily accessible by the general research community.

## 5 EXPERIMENTS

To demonstrate the effectiveness of FGSM adversarial training with fast training methods, we run a number of experiments on MNIST, CIFAR10, and ImageNet benchmarks. All CIFAR10 experiments in this paper are run on a single GeForce RTX 2080ti using the PreAct ResNet18 architecture, and all ImageNet experiments are run on a single machine with four GeForce RTX 2080tis using the ResNet50 architecture (He et al., 2016). Repositories for reproducing all experiments and the corresponding trained model weights are available at `https://github.com/locuslab/fast_adversarial`.

All experiments using FGSM adversarial training in this section are carried out with random initial starting points and step size $\alpha = 1.25\epsilon$ as described in Section 4.1. All PGD adversaries used at evaluation are run with 10 random restarts for 50 iterations (with the same hyperparameters as those used by Shafahi et al. (2019) but further strengthened with random restarts). Speedup with mixed-precision was incorporated with the Apex `amp` package at the `O1` optimization level for ImageNet experiments and `O2` without loss scaling for CIFAR10 experiments.[3]

### 5.1 VERIFIED PERFORMANCE ON MNIST

Since the FGSM attack is known to be significantly weaker than the PGD attack, it is understandable if the reader is still skeptical of the true robustness of the models trained using this method. To demonstrate that FGSM adversarial training confers real robustness to the model, in addition to evaluating against a PGD adversary, we leverage mixed-integer linear programming (MILP) methods from formal verification to calculate the exact robustness of small, but verifiable models (Tjeng

---

[3]Since CIFAR10 did not suffer from loss scaling problems, we found using the `O2` optimization level without loss scaling for mixed-precision arithmetic to be slightly faster.

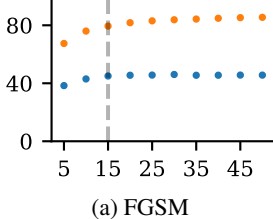 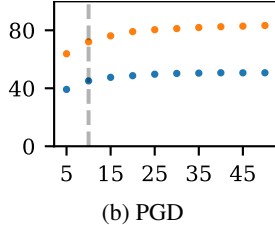 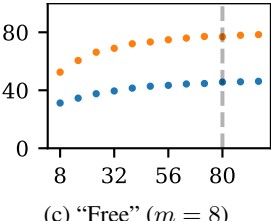

(a) FGSM            (b) PGD            (c) "Free" ($m = 8$)

Figure 2: Performance of models trained on CIFAR10 at $\epsilon = 8/255$ with cyclic learning rates and half precision, given varying numbers of epochs across different adversarial training methods. Each point denotes the average model performance over 3 independent runs, where the $x$ axis denotes the number of epochs $N$ the model was trained for, and the $y$ axis denotes the resulting accuracy. The orange dots measure accuracy on natural images and the blue dots plot the empirical robust accuracy on adversarial images. The vertical dotted line indicates the minimum number of epochs needed to train a model to 45% robust accuracy.

Table 3: Time to train a robust CIFAR10 classifier to 45% robust accuracy using various adversarial training methods with the DAWNBench techniques of cyclic learning rates and mixed-precision arithmetic, showing significant speedups for all forms of adversarial training.

| Method | Epochs | Seconds/epoch | Total time (minutes) |
|---|---|---|---|
| DAWNBench + PGD-7 | 10 | 104.94 | 17.49 |
| DAWNBench + Free ($m = 8$) | 80 | 13.08 | 17.44 |
| DAWNBench + FGSM | 15 | 25.36 | 6.34 |
| PGD-7 (Madry et al., 2017)[5] | 205 | 1456.22 | 4965.71 |
| Free ($m = 8$) (Shafahi et al., 2019)[6] | 205 | 197.77 | 674.39 |

et al., 2017). We train two convolutional networks with 16 and 32 convolutional filters followed by a fully connected layer of 100 units, the same architecture used by Tjeng et al. (2017). We use both PGD and FGSM adversarial training at $\epsilon = 0.3$, where the PGD adversary for training has 40 iterations with step size 0.01 as done by Madry et al. (2017). The exact verification results can be seen in Table 2, where we find that FGSM adversarial training confers empirical and verified robustness which is nearly indistinguishable to that of PGD adversarial training on MNIST.[4]

## 5.2 FAST CIFAR10

We begin our CIFAR10 experiments by combining the DAWNBench improvements from Section 4.2 with various forms of adversarial training. For $N$ epochs, we use a cyclic learning rate that increases linearly from 0 to $\lambda$ over the first $N/2$ epochs, then decreases linearly from $\lambda$ to 0 for the remaining epochs, where $\lambda$ is the maximum learning rate. For each method, we individually tune $\lambda$ to be as large as possible without causing the training loss to diverge, which is the recommended learning rate test from Smith & Topin (2018).

To identify the minimum number of epochs needed for each adversarial training method, we repeatedly run each method over a range of maximum epochs $N$, and then plot the final robustness of each trained model in Figure 2. While all the adversarial training methods benefit greatly from the cyclic learning rate schedule, we find that both FGSM and PGD adversarial training require much fewer epochs than free adversarial training, and consequently reap the greatest speedups.

---

[4]Exact verification results at $\epsilon = 0.3$ for both the FGSM and PGD trained models are not possible since the size of the resulting MILP is too large to be solved in a reasonable amount of time. The same issue also prevents us from verifying networks trained on datasets larger than MNIST, which have to rely on empirical tests for evaluating robustness.

Table 4: Imagenet classifiers trained with adversarial training methods at $\epsilon = 2/255$ and $\epsilon = 4/255$.

| Method | $\epsilon$ | Standard acc. | PGD+1 restart | PGD+10 restarts | Total time (hrs) |
|---|---|---|---|---|---|
| FGSM | 2/255 | 60.90% | 43.46% | 43.43% | 12.14 |
| Free ($m = 4$) | 2/255 | 64.37% | 43.31% | 43.28% | 52.20 |
| FGSM | 4/255 | 55.45% | 30.28% | 30.18% | 12.14 |
| Free ($m = 4$) | 4/255 | 60.42% | 31.22% | 31.08% | 52.20 |

Table 5: Time to train a robust ImageNet classifier using various fast adversarial training methods

| Method | Precision | Epochs | Min/epoch | Total time (hrs) |
|---|---|---|---|---|
| FGSM (phase 1) | single | 6 | 22.65 | 2.27 |
| FGSM (phase 2) | single | 6 | 65.97 | 6.60 |
| FGSM (phase 3) | single | 3 | 114.45 | 5.72 |
| FGSM | single | 15 | - | 14.59 |
| Free ($m = 4$) | single | 92 | 34.04 | 52.20 |
| FGSM (phase 1) | mixed | 6 | 20.07 | 2.01 |
| FGSM (phase 2) | mixed | 6 | 53.39 | 5.34 |
| FGSM (phase 3) | mixed | 3 | 95.93 | 4.80 |
| FGSM | mixed | 15 | - | 12.14 |
| Free ($m = 4$) | mixed | 92 | 25.28 | 38.76 |

Using the minimum number of epochs needed for each training method to reach a baseline of 45% robust accuracy, we report the total training time in Table 3. We find that while all adversarial training methods benefit from the DAWNBench improvements, FGSM adversarial training is the fastest, capable of learning a robust CIFAR10 classifier in 6 minutes using only 15 epochs. Interestingly, we also find that PGD and free adversarial training take comparable amounts of time, largely because free adversarial training does not benefit from the cyclic learning rate as much as PGD or FGSM adversarial training.

## 5.3 Fast ImageNet

Finally, we apply all of the same techniques (FGSM adversarial training, mixed-precision, and cyclic learning rate) on the ImageNet benchmark. In addition, the top submissions from the DAWNBench competition for ImageNet utilize two more improvements on top of this, the first of which is the removal of weight decay regularization from batch normalization layers. The second addition is to progressively resize images during training, starting with larger batches of smaller images in the beginning and moving on to smaller batches of larger images later. Specifically, training is divided into three phases, where phases 1 and 2 use images resized to 160 and 352 pixels respectively, and phase 3 uses the entire image. We train models to be robust at $\epsilon = 2/255$ and $\epsilon = 4/255$ and compare to free adversarial training in Table 4, showing similar levels of robustness. In addition to using ten restarts, we also report the PGD accuracy with one restart to reproduce the evaluation done by Shafahi et al. (2019).

---

[5]Runtimes calculated on our hardware using the publicly available training code at `https://github.com/MadryLab/cifar10_challenge`.

[6]Runtimes calculated on our hardware using the publicly available training code at `https://github.com/ashafahi/free_adv_train`.

With these techniques, we can train an ImageNet classifier using 15 epochs in 12 hours using FGSM adversarial training, taking a fraction of the cost of free adversarial training as shown in Table 5.[7] We compare to the best performing variation of free adversarial training which which uses $m = 4$ minibatch replays over 92 epochs of training (scaled down accordingly to 23 passes over the data). Note that free adversarial training can also be enhanced with mixed-precision arithmetic, which reduces the runtime by 25%, but is still slower than FGSM-based training. Directly combining free adversarial training with the other fast techniques used in FGSM adversarial training for ImageNet results in reduced performance which we describe in Appendix F.

### 5.4 CATASTROPHIC OVERFITTING

While FGSM adversarial training works in the context of this paper, many other researchers have tried and failed to have FGSM adversarial training work. In addition to using a zero initialization or too large of a step size as seen in Table 1, other design decisions (like specific learning rate schedules or numbers of epochs) for the training procedure can also make it more likely for FGSM adversarial training to fail. However, all of these failure modes result in what we call "catastrophic overfitting", where the robust accuracy with respect to a PGD adversarial suddenly and drastically drops to 0% (on the training data). Due to the rapid deterioration of robust performance, these alternative versions of FGSM adversarial training can be salvaged to some degree with a simple early-stopping scheme by measuring PGD accuracy on a small minibatch of training data, and the recovered results for some of these failure modes are shown in Table 1. Catastrophic overfitting and the early-stopping scheme are discussed in more detail in Appendix D.

### 5.5 TAKEAWAYS FROM FGSM ADVERSARIAL TRAINING

While it may be surprising that FGSM adversarial training can result in robustness to full PGD adversarial attacks, this work highlights some empirical hypotheses and takeaways which we describe below.

1. *Adversarial examples need to span the entire threat model.* One of the reasons why FGSM and R+FGSM as done by Tramèr et al. (2017) may have failed is due to the restricted nature of the generated examples: the restricted (or lack of) initialization results in perturbations which perturb each dimension by either $0$ or $\pm\epsilon$, and so adversarial examples with feature perturbations in between are never seen. This is discussed further in Appendix D.

2. *Defenders don't need strong adversaries during training.* This work suggests that rough approximations to the inner optimization problem are sufficient for adversarial training. This is in contrast to the usage of strong adversaries at evaluation time, where it is standard practice to use multiple restarts and a large number of PGD steps.

## 6 CONCLUSION

Our findings show that FGSM adversarial training, when used with random initialization, can in fact be just as effective as the more costly PGD adversarial training. While a single iteration of FGSM adversarial training is double the cost of free adversarial training, it converges significantly faster, especially with a cyclic learning rate schedule. As a result, we are able to learn adversarially robust classifiers for CIFAR10 in minutes and for ImageNet in hours, even faster than free adversarial training but with comparable levels of robustness. We believe that leveraging these significant reductions in time to train robust models will allow future work to iterate even faster, and accelerate research in learning models which are resistant to adversarial attacks. By demonstrating that extremely weak adversarial training is capable of learning robust models, this work also exposes a new potential direction in more rigorously explaining when approximate solutions to the inner optimization problem are sufficient for robust optimization, and when they fail.

---

[7]We use the implementation of free adversarial training for ImageNet publicly available at `https://github.com/mahyarnajibi/FreeAdversarialTraining` and reran it on the our machines to account for any timing discrepancies due to differences in hardware

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

Table 6: Ablation study showing the performance of R+FGSM from Tramèr et al. (2017) and the various changes for the version of FGSM adversarial training done in this paper, over 10 random seeds.

| Method | Step size | Initialization | Robust accuracy |
|---|---|---|---|
| R+FGSM (Tramèr et al., 2017) | 0.15 | Hypercube(0.15) | $34.58 \pm 36.06\%$ |
| R+FGSM (+full step size) | 0.30 | Hypercube(0.15) | $26.53 \pm 32.48\%$ |
| R+FGSM (+uniform init.) | 0.15 | Uniform(0.3) | $72.92 \pm 10.40\%$ |
| Uniform + full (ours) | 0.30 | Uniform(0.3) | $86.21 \pm 00.75\%$ |

Table 7: Training parameters used for the DAWNBench experiments of Table 1

| Parameter | FGSM | PGD | Free |
|---|---|---|---|
| Epochs | 30 | 40 | 96 |
| Max learning rate | 0.2 | 0.2 | 0.04 |

## A  A DIRECT COMPARISON TO R+FGSM FROM TRAMÈR ET AL. (2017)

While a randomized version of FGSM adversarial training was proposed by Tramèr et al. (2017), it was not shown to be as effective as adversarial training against a PGD adversary. Here, we note the two main differences between our approach and that of Tramèr et al. (2017).

1. The random initialization used is different. For a data point $x$, we initialize with the uniform distribution in the entire perturbation region with

$$x' = x + \text{Uniform}(-\epsilon, \epsilon).$$

In comparison, Tramèr et al. (2017) instead initialize on the surface of a hypercube with radius $\epsilon/2$ with

$$x' = x + \frac{\epsilon}{2}\text{Normal}(0, 1).$$

2. The step sizes used for the FGSM step are different. We use a full step size of $\alpha = \epsilon$, whereas Tramèr et al. (2017) use a step size of $\alpha = \epsilon/2$.

To study the effect of these two differences, we run all combinations of either initialization with either step size on MNIST. The results are summarized in Table 6.

We find that using a uniform initialization adds the greatest marginal improvement to the original R+FGSM attack, while using a full step size doesn't seem to help on its own. Implementing both of these improvements results in the form of FGSM adversarial training presented in this paper. Additionally, note that R+FGSM as done by Tramèr et al. (2017) has high variance in robust performance when done over multiple random seeds, whereas our version of FGSM adversarial training is significantly more consistent and has a very low standard deviation over random seeds.

## B  TRAINING PARAMETERS FOR TABLE 1

For all methods, we use a batch size of 128, and SGD optimizer with momentum 0.9 and weight decay $5 * 10^{-4}$. We report the average results over 3 random seeds. The remaining parameters for learning rate schedules and number of epochs for the DAWNBench experiments are in Table 7. For runs using early-stopping, we use a 5-step PGD adversary with 1 restart on 1 training minibatch to detect overfitting to the FGSM adversaries, as described in more detail in Section D.

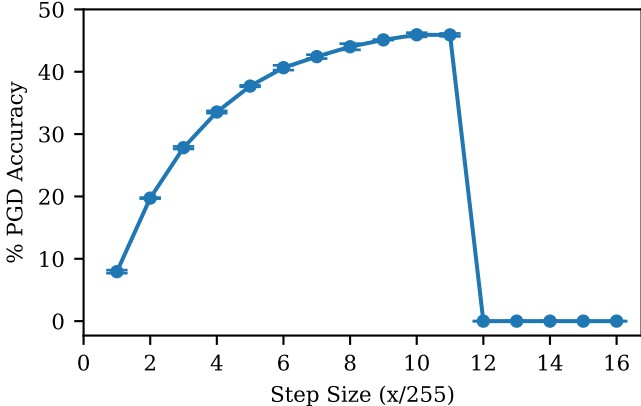

Figure 3: Robust test performance of FGSM adversarial training over different step sizes for $\epsilon = 8/255$.

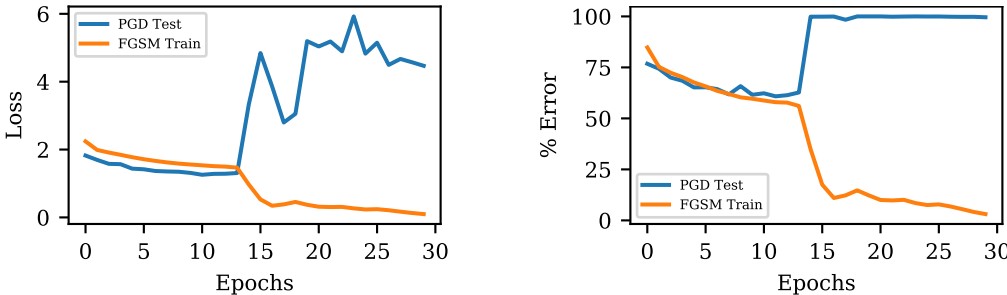

Figure 4: Learning curves for FGSM adversarial training plotting the training loss and error rates incurred by an FGSM and PGD adversary when trained with zero-initialization FGSM at $\epsilon = 8/255$, depicting the catastrophic overfitting where PGD performance suddenly degrades while the model overfits to the FGSM performance.

## C    OPTIMAL STEP SIZE FOR FGSM ADVERSARIAL TRAINING

Here, we test the effect of step size on the performance of FGSM adversarial training. We plot the mean and standard error of the robust accuracy for models trained for 30 epochs over 3 random seeds in Figure 3, and vary the step size from $\alpha = 1/255$ to $\alpha = 16/255$.

We find that we get increasing robust performance as we increase the step size up to $\alpha = 10/255$. Beyond this, we see no further benefit, or find that the model is prone to overfitting to the adversarial examples, since the large step size forces the model to overfit to the boundary of the perturbation region.

## D    CATASTROPHIC OVERFITTING AND THE EFFECT OF EARLY STOPPING

While the main experiments in this paper work as is (with the cyclic learning rate and FGSM adversarial training with uniform random initialization), many of the variations of FGSM adversarial training which have been found to not succeed all fail similarly: the model will very rapidly (over the span of a couple epochs) appear to overfit to the FGSM adversarial examples. What was previously a reasonably robust model will quickly transform into a non-robust model which suffers 0% robust accuracy (with respect to a PGD adversary). This phenomenon, which we call catastrophic overfitting, can be seen in Figure 4 which plots the learning curves for standard, vanilla FGSM adversarial training from zero-initialization.

Indeed, one of the reasons for this failure may lie in the lack of diversity in adversarial examples generated by these FGSM adversaries. For example, using a zero initialization or using the random

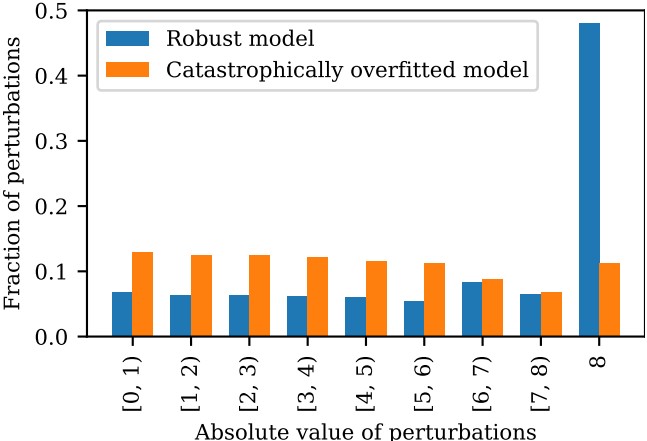

Figure 5: Histogram of the resulting perturbations from a PGD adversary for each feature for a successfully trained robust CIFAR10 model and a catastrophically overfitted CIFAR10 model.

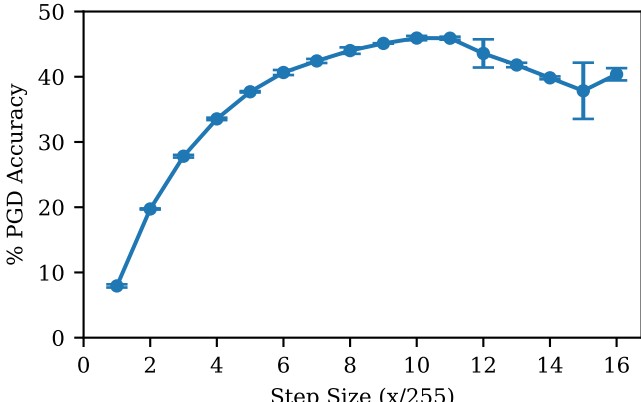

Figure 6: Robust test performance of FGSM adversarial training over different step sizes for $\epsilon = 8/255$ with early stopping to avoid catastrophic overfitting.

initialization scheme from Tramèr et al. (2017) will result in adversarial examples whose features have been perturbed by $\{-\epsilon, 0, \epsilon\}$, and so the network learns a decision boundary which is robust only at these perturbation values. This can be verified by running a PGD adversarial attack on models which have catastrophically overfitted, where the perturbations tend to be more in between the origin and the boundary of the threat model (relative to a non-overfitted model, which tends to have perturbations near the boundary), as seen in Figure 5.

These failure modes, including the other failure modes discussed in Section 5.4, can be easily detected by evaluating the PGD performance on a small subset of the training data, as the catastrophic failure will result in 0% robust accuracy for a PGD adversary on the training set. In practice, we find that this can be a simple as a single minibatch with a 5-step PGD adversary, which can be quickly checked at the end of the epoch. If robust accuracy with respect to this adversary suddenly drops, then we have catastrophic overfitting. Using a PGD adversary on a training minibatch to detect catastrophic overfitting, we can early stop to avoid catastrophic overfitting and achieve a reasonable amount of robust performance.

For a concrete example, recall from Section C that step sizes larger than $11/255$ result in 0% robust accuracy, due to this catastrophic overfitting phenomenon. By using early stopping to catch the model at its peak performance before overfitting, FGSM adversarial training with larger step sizes can actually achieve some degree of robust accuracy, as shown in Figure 6.

Table 8: Training parameters used for Figure 2

| Parameter | FGSM | PGD | Free |
|---|---|---|---|
| Max learning rate | 0.2 | 0.2 | 0.04 |

Table 9: ImageNet classifiers trained with free adversarial training methods at $m = 3$ minibatch replay when augmented with DAWNBench optimizations, against $\ell_\infty$ perturbations of radius $\epsilon = 4/255$, where 30 epochs of free training is equivalent to 15 epochs of FGSM training

| Method | Step size | Epochs | Standard acc. | PGD+1 | PGD+10 |
|---|---|---|---|---|---|
| Free+DAWNBench | 4/255 | 15 | 49.87% | 22.78% | 22.18% |
| Free+DAWNBench | 5/255 | 15 | 50.48% | 22.88% | 22.25% |
| Free+DAWNBench | 4/255 | 30 | 49.87% | 28.17% | 27.08% |
| Free+DAWNBench | 5/255 | 30 | 50.48% | 28.73% | 27.81% |
| Free ($m = 4$) | 4/255 | 92 | 60.42% | 31.22% | 31.08% |
| FGSM | 5/255 | 15 | 55.45% | 30.28% | 30.18% |

## E  TRAINING PARAMETERS FOR FIGURE 2

For all methods, we use a batch size of $128$, and SGD optimizer with momentum $0.9$ and weight decay $5 * 10^{-4}$. We report the average results over 3 random seeds. Maximum learning rates used for the cyclic learning rate schedule are shown in Table 8.

## F  COMBINING FREE ADVERSARIAL TRAINING WITH DAWNBENCH IMPROVEMENTS ON IMAGENET

While adding mixed-precision is a direct speedup to free adversarial training without hurting performance, using other optimization tricks such as the cyclic learning rate schedule, progressive resizing, and batch-norm regularization may affect the final performance of free adversarial training. Since ImageNet is too large to run a comprehensive search over the various parameters as was done for CIFAR10 in Table 3, we instead test the performance of free adversarial training when used as a drop-in replacement for FGSM adversarial training with all the same optimizations used for FGSM adversarial training. We use free adversarial training with $m = 3$ minibatch-replay, with 2 epochs for phase one, 2 epochs for phase two, and 1 epoch for phase three to be equivalent to 15 epochs of standard training. PGD+$N$ denotes the accuracy under a PGD adversary with $N$ restarts.

A word of caution: this is not to claim that free adversarial training is completely incompatible with the DAWNBench optimizations on ImageNet. By giving free adversarial training more epochs, it may be possible recover the same or better performance. However, tuning the DAWNBench techniques to be optimal for free adversarial training is not the objective of this paper, and so this is merely to show what happens if we naively apply the same DAWNBench tricks used for FGSM adversarial training to free adversarial training. Since free adversarial training requires more epochs even when tuned with DAWNBench improvements for CIFAR10, we suspect that the same behavior occurs here for ImageNet, and so 15 epochs is likely not enough to obtain top performance for free adversarial training. Since one epoch of FGSM adversarial training is equivalent to two epochs of free training, a fairer comparison is to give free adversarial training 30 epochs instead of 15. Even with double the epochs (and thus the same compute time as FGSM adversarial training), we find that it gets closer but doesn't quite recover the original performance of free adversarial training.

