# OpenReview forum: "Fast is better than free: Revisiting adversarial training"
_ICLR.cc/2020/Conference — Accept (Poster)_

### Official Review · AnonReviewer1 · 2019-10-17
**Official Blind Review #1**

**Rating:** 6

**Review:**

This paper revisits Random+FGSM method to train robust models against strong PGD evasion attacks. Coupled together with tricks for accelerating natural training, such as cyclic learning rate, mixed precision, the robust models can be trained faster than previous methods.

+The experimental results are impressive. The trained model is robust (at Madry’s PGD level), and the total training procedure is fast (6 min for CIFAR-10 and 12 hr for ImageNet).

+ The method is simple, and I guess reproducible.

+The paper shows surprising facts of a well-known method.

+The paper is generally well-written and easy to follow.

I do have some concerns of the work
- The paper is empirical and the techniques are combinations of previous methods. Even for the surprising fact that Random+FGSM, it has been discussed in several previous papers, for example,  ICLR 2019 Defensive Quantization: When Efficiency Meets Robustness  https://openreview.net/forum?id=ryetZ20ctX. So the main contribution of the paper is limited to show RFGSM works well when combined with optimization tricks like cyclic learning rate.

-In previous methods claiming random+FGSM can train robust model, their method seems to be slightly different from Alg 3 in page 4 of this paper. The alg in this paper seems to be identical to Madry’s implementation of R-FGSM, which is shown not robust to PGD attacks. See discussions in https://openreview.net/forum?id=rJzIBfZAb and https://openreview.net/forum?id=ryetZ20ctX. I would like the authors to clarify their method to resolve such conflicts and make it clear how R-FGSM can be as robust as PGD as in table 1.

-The first two paragraphs of section 4.1 seem to be inaccurate. One important trick in the “adversarial training for free” paper is to replay each minibatch m times. It is hard to say how much nonzero initialization helps. According to “universal adversarial training” (https://arxiv.org/pdf/1811.11304.pdf). It may help, but cannot compete with Madry’s PGD training when defending against PGD attacks.

-I am not sure if using a larger norm 1.25 * \epsilon is a fair comparison. A baseline of PGD training bounded by 1.25 * \epsilon would help.

-Could the authors combine table 4 and 5 for easy comparison of robust accuracy and training time? Did the authors try the optimization tricks on ImageNet for the baseline free adversarial training method?



================== after rebuttal =================
I change my rating to weak accept. I tend to accept for the following reason
(1) there seems to be no obvious flaw in the authors implementation. I quickly skimmed their code, and looks like a few researchers have tried their code and responded in public discussion. The surprising robustness of RFGSM, though the originality is questionable and the technical difference comparing to previous methods are subtle, seems to hold true.
(2) The authors work hard to address the comments.
I still have some concerns, mainly regarding the fairness of experimental comparison.
(1) As pointed out in public discussion, the success of the proposed RFGSM relies on early stopping. I am wondering if early stopping also helps other methods since it turns out to be some sort of selection procedure.
(2) The authors did not update time in table 1 for CIFAR-10 results, which I consider almost no extra efforts. I am wondering how much more time each method needs from 45% in figure 2 to higher robust accuracy in table 1.
(3) I cannot understand why the proposed method is a particular good fit with cyclic LR and low precision tricks comparing to other methods.

**Experience Assessment:**

I have published one or two papers in this area.

**Review Assessment: Checking Correctness Of Derivations And Theory:**

N/A

**Review Assessment: Checking Correctness Of Experiments:**

I assessed the sensibility of the experiments.

**Review Assessment: Thoroughness In Paper Reading:**

I read the paper thoroughly.

---

> ### Author Response · Authors · 2019-11-05
> **Thank you for your review**
>
> Thanks for your feedback regarding the connections to other randomized FGSM methods. This topic has already occurred in the public discussion, and so our response will largely reflect that. We will first discuss the main differentiating factors between our approach and the previous R+FGSM approach by Tramer et al., and follow up by answering the remaining comments.
>
> R+FGSM:
> Indeed, there has been previous work on using randomization with FGSM (e.g. R+FGSM as done by Tramer et al., which is the one used in "Defensive Quantization: When Efficiency Meets Robustness"). We note, however, that the R+FGSM approach from Tramer et al. is also the same randomized FGSM approach considered by Madry et al., which considers both the vanilla, non-randomized attack (which they conclude is not robust) as well as the R+FGSM attack from Tramer et al. as mentioned in their paper in Table 5 on page 17 of the Appendix.
>
> Our approach at using randomization with FGSM is very similar to Tramer et al. but differs in two aspects which are quite critical to the consistency and effectiveness of the defense. In fact, a lot of this discussion has already occurred in the public comments below (e.g. see our discussion with Florian here, where we explicitly compare our approach with R+FGSM: https://openreview.net/forum?id=BJx040EFvH&noteId=HJe2trIsDS), but we can summarize the main points for your convenience: namely, by using 1) a different random initialization and 2) a larger step size, our version of randomized FGSM adversarial training converges to better defended models with much higher consistency. By rerunning the approaches multiple times with different random seeds, one gets a fuller picture: R+FGSM as done by Tramer et al. has high variance and worse performance with respect to random seeds, whereas our approach consistent achieves results comparable to PGD adversarial training regardless of random seed (see the table at  https://github.com/anonymous-sushi-armadillo/openreview/blob/master/README.md which we generated for the referenced public discussion). So we believe that the contribution of this paper extends beyond just incorporating DAWNBench speedups.
>
> On non-zero initialization and the connection to Free Adversarial Training:
> The usage of minibatch-replay in free adversarial training is indeed the second key difference between free and FGSM adversarial training. While we don't mention this at the start of section 4.1, we do discuss this difference later in the last paragraph of the same section. However, we focused primarily on the initialization, because in our experiment in Table 1, we show the effect of using various initializations for FGSM adversarial training without changing any other parameters: simply going from zero to non-zero results in large gains in robustness. Note that the Universal Adversarial Training paper also uses R+FGSM as done by Tramer et al., and so it suffers from the same drawbacks as described above.
>
> On the "larger norm" and fair comparison:
> We believe there may be a misunderstanding here: the model is trained against an adversary bounded by epsilon, but the alpha=1.25*epsilon is merely the step size for the FGSM attack. Indeed, regardless of the step size, the FGSM attack is still projected back to the epsilon boundary. As a PGD adversary is allowed to tune the step size and the number of steps it takes to find an adversarial example, it should also be fair for the FGSM adversary to also tune its singular step size, as both methods project onto the same radius ball.
>
> Other comments:
> Thank you for your suggestion, yes we can certainly add the training times to Table 4 to make it easier to connect the two.
>
> We primarily focused on optimizing the DAWNBench improvements with Free adversarial training in the CIFAR10 setting, since the problem setting allows for extensive tuning of all parameters (which is not as feasible in the ImageNet setting).

---

> > ### Comment · AnonReviewer1 · 2019-11-07
> > **Thanks for clarification, more experiments help**
> >
> > Thanks for the clarification on larger norm and fair comparison. Correct me if I was wrong, the paper claims two major contributions
> > (1) Showing R-FGSM can be as robust as PGD for training
> > (2) Train robust model faster with R-FGSM and optimization tricks
> >
> > I also read the open reviews and have a quick look of the authors' code.  The authors have convinced me on the (1) contribution. I am now on the fence.
> >
> > Since this paper has attracted quite some attention, I may have potentially set a higher standard for it.  I would be happy to change my score if the authors are willing to try harder to convince me on the (2) contribution.
> > (I) Gradient clipping nn.utils.clip_grad_norm_(model.parameters(), 0.5) seems to be unnecessary for ResNet on CIFAR. Could the authors clarify?
> > (II) It makes sense that 2*\eps stepsize does not work quite well. But how sensitive is the stepsize between \eps and 2*\eps, and what is the sweet point? I hope the authors could provide more ablation study. On CIFAR is good enough. It would also help to do multiple runs and show variance. It could be done during rebuttal period since each run only takes 6 min.
> > (III) I strongly encourage the authors to include training time in table 1 and table 4 for directly comparison.
> > (IV) I would hope the authors clarify results on Free method, and maybe add some experiments. I will use adversarial stepsize for updating perturbation, and learning rate (LR) for updating model weights.  What is the adversarial stepsize in Free when combined with DAWNBench optimization tricks? Did you use the same LR and LR scheduler and other non-adversarial-related setting for Free and RFGSM, and natural training? I would like to see results on ImageNet for Free with DAWNBench tricks, for adversarial stepsize \eps and 1.25*\eps, you can cut it off at 15 epochs to compare with RFGSM. Each run should take about 7 hours and can be finished in rebuttal period.
> >
> > I will give it extra points if you can try sanity check with CW attacks.

---

> > > ### Author Response · Authors · 2019-11-08
> > > **Some initial answers**
> > >
> > > Thanks for following up! We're of course glad to continue the discussion. A couple of your questions can be quickly answered, so we'll begin by answering those, and follow up with the remaining questions later.
> > >
> > > (I) Gradient clipping is unnecessary most likely because the cyclic learning rate schedule from DAWNBench already scales the gradients. A cyclic learning rate schedule peaks when the training loss begins to diverge (this point can be found with the learning rate test mentioned in Section 5.2 of the paper, which comes directly from Smith & Topin 2018). As a result, the model gradients throughout the training procedure, when scaled by the learning rate, are quite stable and so gradient clipping is unnecessary (which is typically used to combat exploding gradients and loss divergence in the training procedure).
> > >
> > > (II) This was a question that also came up during the public discussion, and so we can conveniently already answer this for you. You can find a plot of runs over varying step sizes here (which, of course, we intend on adding to the paper):
> > >
> > > https://github.com/anonymous-sushi-armadillo/openreview/blob/master/step_size_cifar10.pdf
> > >
> > > (III) Yes, we agree that this is a good idea, and the change will be present once we've uploaded a revised version of the paper.
> > >
> > > The remaining parts will have to wait, but we'll be running the experiments and will follow up when they are done.

---

> > > > ### Comment · AnonReviewer1 · 2019-11-13
> > > > **Thanks for the pointers**
> > > >
> > > > Thanks for the clarification.
> > > >
> > > > (I) I thought the authors used gradient clipping in their code, but it is probably not common approach for CIFAR benchmark? line 210 & 222 here https://github.com/anonymous-sushi-armadillo/fast_is_better_than_free_CIFAR10/blob/master/train_cifar.py
> > > >
> > > > (III) Could you add time to table 1?

---

> > > > > ### Author Response · Authors · 2019-11-14
> > > > > **A few more pointers**
> > > > >
> > > > > (I) We apologize for the confusion. Gradient clipping is unnecessary, and the code works perfectly well without it (it is a redundant artifact of the submitted code from experimenting, and we forgot to remove it from the CIFAR repository before creating the anonymous repository). Just to be safe, since the CIFAR10 experiments are quite fast, we double checked and reran the code to confirmed that the results are still consistent without the gradient clipping (note that the submitted code for MNIST and ImageNet both correctly don't use gradient clipping). With both cyclic learning rates and automatic loss scaling from the AMP mixed-precision arithmetic (both of which help combat exploding gradients), clipping gradients is indeed redundant for avoiding exploding gradients. Thank you for the taking such a detailed look at the code and pointing this out!
> > > > >
> > > > > (III) Yes, we can certainly add this.
> > > > >
> > > > > Thank you once again for your valuable, detailed feedback!

---

> > > ### Author Response · Authors · 2019-11-11
> > > **Following up on the remaining points**
> > >
> > > In this post we address the two remaining points that required some additional experiments to be run.
> > >
> > > (IV) As the reviewer suggested, we ran the Free adversarial training approach with the same LR schedule and training procedure as used for FGSM adversarial training on ImageNet, for both stepsizes \alpha=\epsilon and \alpha=1.25*\epsilon, for \epsilon=4/255 over 15 epochs. In short, this results in (against a 50 step PGD adversary with 10 restarts)
> > >
> > > Free training with \alpha=\epsilon: 22.18% robust accuracy
> > > Free training with \alpha=\epsilon*1.25: 22.25% robust accuracy
> > >
> > > For reference, in comparison, our FGSM adversarial training achieves 30.18% robust accuracy in the same setting with the same number of epochs. Note that these numbers for Free adversarial training are lower than what Free adversarial training can get with many more epochs, and so we see a similar trend as in the CIFAR10 experiments from Table 3: even though DAWNBench can speed up all the methods, the simultaneous gradient updates in Free adversarial training method ends up requiring more epochs to achieve the best possible performance.
> > >
> > >
> > > (V) Regarding CW attacks: we've done our best to reproduce the CW attack for the L-infinity perturbation model, relying on the implementation by Nicholas Carlini here: https://github.com/carlini/nn_robust_attacks/blob/master/li_attack.py
> > >
> > > Note that we had to adjust the factors for increasing/decreasing various constants in order for the runtime to even be feasible. With the default parameters in the reference implementation, it can take up to 704k iterations to perform one single attack, which is not feasible for our hardware (and takes about 6 hours to run one attack). We adjusted the constants to use in total 2k gradient iterations per attack, which is still far beyond the 50-step PGD adversary used in the paper.
> > >
> > > On our CIFAR10 model trained with FGSM adversarial training, the attack achieves 54.49% robust accuracy, in comparison to the 46.25% robust accuracy when evaluated with the PGD based adversary, so no drastic changes here.
> > >
> > > One final note: while it's not generally a bad idea to run multiple attacks when applicable, it is important to understand the purpose behind running additional attacks. Throughout the literature for the L-infinity threat model, the PGD based attack has generally outperformed the CW attack while being much more efficient, and this is why we initially did not consider running it. In fact, you'll find that the open source adversarial attack libraries (e.g. Cleverhans for Tensorflow and Foolbox for PyTorch) do not implement the CW attack for L-infinity perturbations (it is only implemented for L2 perturbations). In the case of Cleverhans, the author for the CW attack recognized this, and decided to not add it to the library for this reason, which you can see in this issue here: https://github.com/tensorflow/cleverhans/issues/978

---

> > > > ### Comment · AnonReviewer1 · 2019-11-13
> > > > **More clarification**
> > > >
> > > > I thank the authors for their efforts.
> > > >
> > > > (IV) I hope two clarification from the authors: with same number of 15 epochs, Free method is half the time of proposed RFGSM, right? Why use m=4 for Free method. The original paper often uses m=8.
> > > >
> > > > (V) I appreciate the authors working on it. I said it is considered a plus. The cross-entropy loss used in PGD attack can lead to strange behavior related to gradient masking. I just want one more sanity check.  You could also try black-box attack, or run PGD for 1000 steps, which I personally consider too much.

---

> > > > > ### Author Response · Authors · 2019-11-14
> > > > > **More answers**
> > > > >
> > > > > Thanks again for following up!
> > > > >
> > > > > (IV) Yes, this is correct: one epoch of free training takes half the time of randomized FGSM since it does 1 backwards pass instead of 2. We noticed this with your initial request for 15 epochs, and so we've actually already started the corresponding experiments for 30 epochs of free training instead of 15 (in order to be more fair and have comparable compute times). We will of course update the paper with these results once they are done.
> > > > >
> > > > > To answer your second question, we use m=4 for ImageNet specifically because in the Free paper, they found that m=4 performed the best *for ImageNet* (it achieves about 3-4% more PGD accuracy than m=8). You can see this in Tables 3 and 7 in the free adversarial training paper. The usage of m=8 is the optimal minibatch replay value for free training on CIFAR10, and not ImageNet. Throughout the paper we compare to the *best* hyperparameter for minibatch replay for each specific dataset, which is why we compare to m=8 for CIFAR10 and m=4 for ImageNet, so that we do not unfairly cripple the free adversarial training benchmark. Note that this highlights another advantage of using FGSM adversarial training: there is no need to tune a minibatch replay parameter.
> > > > >
> > > > > (V) We ran the PGD attack on our FGSM trained CIFAR10 model for 1000 iterations as requested (also with 10 random restarts, in order for the adversary to be strictly stronger than what we previously considered), and in comparison to the 50 iteration PGD adversary, we observe that the stronger adversary results in a drop of 0.21% robust accuracy (from 46.46% to 46.25% for a single CIFAR10 model).

---

### Official Review · AnonReviewer2 · 2019-10-22
**Official Blind Review #2**

**Rating:** 6

**Review:**

The main claim of this paper is that a simple strategy of randomization plus fast gradient sign method (FGSM) adversarial training yields robust neural networks. This is somewhat surprising because previous works indicate that FGSM is not a powerful attack compared to iterative versions of it like projected gradient descent (PGD), and it has not been shown before that models trained on FGSM can defend against PGD attacks. Judging from the results in the paper alone, there are some issues with the experiment results that could be due to bugs or other unexplained experiment settings.

The most alarming part of the results is the catastrophic failure with larger step sizes 16/255 for CIFAR10 in Table 1. This is very strange because the method works well when using epsilon=10/255 to defend against an adversary with epsilon=8/255.
The authors explain this with overfitting, but this is not satisfactory. Suppose I want to use the method to defend against an adversary with power epsilon=14/255, then it is conceivable that I would use a slightly larger step size, say 16/255, as suggested by the authors.  The results in the table tells me that this method will fail completely, because it cannot defend against epsilon=8/255, let alone the target perturbation 14/255. The method is probably not failing completely, because it does have good accuracy on clean data and does learn something. So it cannot be due to the model not having enough capacity to learn against an epsilon=16/255 adversary.

The authors should check some potential issues with the experiments:
1. Is there any label leakage in the FGSM training?
2. The pseudo-code does not contain any projection onto the feasible set; the authors should check it.

Since the claim of this paper is somewhat unexpected given previous works on defending against adversaries, the experiment results have to be very solid. With these issues with the experiments I don't believe the current paper is ready for publication yet.

After rebuttal:
The authors' new experiments and response answers most of my concerns.


**Experience Assessment:**

I have published one or two papers in this area.

**Review Assessment: Checking Correctness Of Derivations And Theory:**

I assessed the sensibility of the derivations and theory.

**Review Assessment: Checking Correctness Of Experiments:**

I assessed the sensibility of the experiments.

**Review Assessment: Thoroughness In Paper Reading:**

I read the paper thoroughly.

---

> ### Author Response · Authors · 2019-11-05
> **Clarification of the given example and addressing the potential issues**
>
> Thanks for your feedback. With regards to the catastrophic overfitting observed at larger step sizes, we first clarify a misunderstanding here: defending against an adversary with radius epsilon means that we project the perturbation onto the ball with radius epsilon. With regards to your example, indeed, a step size of 16/255, *when projected onto a ball of radius 8/255*, results in overfitting, as large step size forces the generated adversarial examples to be clustered at the boundary.
>
> However, if, as you describe, we instead want to defend against an adversary with radius 14/255 using a step size of 16/255, then note that we project the FGSM step on the ball of radius 14/255. This is a fundamentally different scenario from the results in the table, which project onto a ball of radius 8/255, and so the table does not imply that the method is guaranteed to fail. Indeed, since the projected radius is larger, the adversarial examples are not clustered at the boundary, and so there is no overfitting.
>
> In short: a large step size of 16/255 fails when projected onto a radius of 8/255, but works perfectly well when projected onto a similarly large radius (e.g. 14/255). This is why we wrote alpha=1.25*epsilon.
>
> As for the potential issues in the experiments, we note below that they are not at all issues, and hope that the reviewer can reconsider:
>
> Label leaking:
> We do not observe label leaking (as defined in "Adversarial Machine Learning at Scale" by Kurakin et al. 2017). You can see this in all of our experimental results, e.g. in Table 1, Table 2, Table 4, and Figure 2, where the standard accuracy always is above the adversarial accuracy, and this behavior can be verified in the models that we have released.
>
> Projection in pseudocode:
> The projection is in fact present in our pseudocode. It is the line that says \delta = max(min(\delta,\epsilon), -\epsilon). It is also in our submitted code. If you are referring to clipping at the [0,255] bounds from the image, this is also done in our code (as described in the public discussion).

---

### Official Review · AnonReviewer3 · 2019-10-23
**Official Blind Review #3**

**Rating:** 8

**Review:**

The authors claimed a classic adversarial training method, FGSM with random start, can indeed train a model that is robust to strong PGD attacks. Moreover, when it is combined with some fast  training methods, such as cyclic learning rate scheduling and mixed precision, the adversarial training time can be significantly decreased. The experiment verifies the authors' claim convincingly.
Overall, the paper provides a novel finding that could significantly change the adversarial training strategy. The paper is clearly written and easy to follow. I recommend the acceptance.


**Experience Assessment:**

I have read many papers in this area.

**Review Assessment: Checking Correctness Of Derivations And Theory:**

I assessed the sensibility of the derivations and theory.

**Review Assessment: Checking Correctness Of Experiments:**

I carefully checked the experiments.

**Review Assessment: Thoroughness In Paper Reading:**

I read the paper at least twice and used my best judgement in assessing the paper.

---

> ### Author Response · Authors · 2019-11-05
> **Thanks for your review**
>
> Thanks for your review. Indeed, we hope this this work inspires new analysis which can perhaps quantify the degree to which the inner maximization must be solved in order to perform robust optimization.

---

### Public Comment · ~Florian_Tramer1 · 2019-09-27
**Similarity to R+FGSM in Ensemble Adversarial Training**

Very interesting work!

The proposed FGSM with random initialization is very similar to the R+FGSM attack we had discussed in our paper "Ensemble Adversarial Training: Attacks and Defenses" two years ago: https://openreview.net/forum?id=rkZvSe-RZ

What surprises me here is that we had tried doing adversarial training with the R+FGSM attack on MNIST, but did not find it to be effective. Quoting from our paper:

"We also tried adversarial training using R+FGSM on MNIST, using a similar approach as (Madry et al., 2017).
We adversarially train a CNN (model A in Table 5) for 100 epochs, and attain > 90.0% accuracy on R+FGSM samples. However, training on R+FGSM provides only little robustness to iterative attacks. For the PGD attack of (Madry et al., 2017) with 20 steps, the model attains 18.0% accuracy."

The reason our experiment failed while yours presumably succeed might be related to your discussion on step-size selection (page 5). In our experiments, we were taking a random step of size eps/2 followed by a FGSM step of size eps/2, which in hindsight was too weak to properly explore the l-norm ball.

It would be interesting to include further discussion or experiments on the effect (and brittleness) of the step-size selection in the paper.

---

> ### Public Comment · ~Anthony_Wittmer1 · 2019-09-27
> **We also failed.**
>
> It's unbelievable that FGSM adversarial training with random initialization can be as effective as PGD adversarial training.
>
> We have tried the experiments of R+FGSM adversarial training, but the robustness is not competitive with that of PGD adversarial training.  Maybe the authors can consider various attacks to evaluate the robustness, such as the black-box attacks.
>
> From the evaluation in Tabel 1, the performace of the proposed method seems to be very sensitive  to the step size.

---

> ### Author Response · Authors · 2019-09-27
> **Thanks for your comments**
>
> Hi everyone,
> Good to see this paper is already generating controversy! (We expected as much :) )
>
> First off, thanks Florian for pointing out the connection to the R+FGSM method you tried.  We know the paper and ensembling technique well, of course, but honestly that connection slipped our mind as it was considered a failed option in that paper, so wasn't the focus.  We'll definitely add this connection and discussion.  As we hope is clear, our goal here is definitely not to claim a new algorithm, but just that this old approach works surprisingly well when tuned properly (really surprising to us too).
>
> You're absolutely right that the step size has some effect here, and it's a great idea to compare this more formally. FWIW, it's not that we just decided alpha=10/255 randomly, but it seemed like slightly (1.25x) larger than the epsilon ball (but notably not 2x, which would be a "full" FGSM step) worked best, and this was consistent across all datasets.  However, there is a reasonable range of choices here that works ok, and we'll add a figure within the next few days (at least for CIFAR) showing the whole curve.
>
> Thanks for the suggestion!

---

> > ### Author Response · Authors · 2019-09-29
> > **Followup comparison to R+FGSM and effect of step sizes**
> >
> > Hi Florian,
> >
> > To shed some light on why the R+FGSM adversarial training that you had tried before wasn't as successful, we conducted some basic experiments. We also show the effects of step size at the end of this post.
> > The main differences between R+FGSM and our approach are the following:
> >
> > (1) The R+FGSM initialization is on the surface of an alpha=epsilon/2 box (e.g. alpha*sign(Normal(0,1))), whereas ours is initialized with Uniform(-epsilon,epsilon).
> >
> > (2) The FGSM step is taken with step size epsilon-alpha=epsilon/2.
> >
> > So we tested what happened when took R+FGSM into our training procedure and changed either (1) or (2) to see why ours succeeded. We put a table of the outcome of this experiment here (https://github.com/anonymous-sushi-armadillo/openreview/blob/master/README.md), where the mean and standard deviation are taken over 10 random seeds. We found R+FGSM to be highly dependent on the random seed: the performance ranged the entire spectrum, from occasionally succeeding to completely failing. By changing the initialization to be Uniform, the variance is greatly reduced and the performance on average is better. However, only changing the step size to be a full epsilon step instead of an alpha/2 step did not seem to help on its own. When these changes are taken in combination (which results in the method we use in the paper), we get the same consistent result with little variance. Hopefully this sheds some light on why the R+FGSM method was unsuccessful while this one was!
> >
> > Additionally, as requested, we have run the CIFAR10 training with varying step sizes to show the effect on FGSM training. The plot is available here (https://github.com/anonymous-sushi-armadillo/openreview/blob/master/step_size_cifar10.pdf). In summary, we find that performance deteriorates with step sizes smaller than epsilon (likely because the adversary is effectively weakened), and that once epsilon is too large (11/255 or higher in this case) it becomes easy to fail (possibly because forcing the adversarial example to the boundary of the L-inf ball makes it easy for the model to overfit).

---

### Public Comment · ~Anthony_Wittmer1 · 2019-09-28
**Evaluation on the black-box attacks**

The finding of this paper is very intriguing. However, the disccusion of the robustness on the black-box attacks is missing in this paper.

---

> ### Author Response · Authors · 2019-09-30
> **On black-box attacks**
>
> Hi Anthony,
>
> While in general it's a good idea to use black box attacks to get around gradient obfuscation defenses, it's a little odd to use them in this setting. Afterall, we are using adversarial training on standard models with standard data preprocessing techniques, and there are no steps which would hide the actual gradient, which is why PGD adversarial training (and consequently, FGSM adversarial training) is not considered to be a defense which obfuscates gradients. Additionally, they are besides the point of the paper: which is to show that FGSM adversarial training can lead to robustness against full strength multi-step PGD adversaries.
>
> That being said, it is fair to say that we could still run the black box attacks anyways. However, it would only be useful as a comparison to the same attacks performed against PGD adversarial trained models, both of which would likely not perform as well as the white-box attacks anyways since clean model gradients are available.

---

> > ### Public Comment · ~Anthony_Wittmer1 · 2019-09-30
> > **On evaluating adversarial robustness**
> >
> > Sorry, I do not agree that it's a little odd to use black-box attack in this setting.
> >
> > When proposing a new adversarial defense, the goal is to produce a model that is robust against _all_ possible attacks within a threat model. For instance, increased robustness to FGSM attacks does not constitute progress if the model is vulnerable to PGD attacks. This is why the robust accuracy of a model is defined as the *minimum* accuracy achieved against the worst-case attack within the threat model.

---

### Public Comment · ~Anthony_Wittmer1 · 2019-09-29
**Pytorch version and the requirements**

Could you tell me the pytorch version and the requirements for the repo fast_is_better_than_free_CIFAR10 ? I failed to load the released model with pytorch 0.4.0 .

Thanks.

---

> ### Author Response · Authors · 2019-09-29
> **Code requirements**
>
> For those who are wishing to run the provided code, we have added more detailed installation instructions and requirements in the README of the CIFAR10 repository: https://github.com/anonymous-sushi-armadillo/fast_is_better_than_free_CIFAR10 .

---

> > ### Public Comment · ~Anthony_Wittmer1 · 2019-09-30
> > **Thanks.**
> >
> > Thanks for providing more details.

---

### Public Comment · ~Dinghuai_Zhang1 · 2019-09-29
**Cannot get the same results, maybe I took something wrong?**

It seems that the codes offered by authors need particular benchmark thus I cannot run them. So I do FGSM with uniform init using my own adv code base. I cannot get consistent results. I think maybe I miss something or get something wrong?

For cifar10, I use resnet18, init lr 0.1, piece-wise decay at 70, 90, 100 epoch and get final PGD20 accuracy is **0.00%** when step size alpha = 10/255.  Other settings follow (Madry et al., 2017)'s paper. My code base is here: https://github.com/a1600012888/YOPO-You-Only-Propagate-Once

---

> ### Author Response · Authors · 2019-09-30
> **Piecewise learning rate with many epochs is another potential failure mode**
>
> Hi Dinghuai,
>
> Regarding running the code, we've added more detailed instructions in the README of the CIFAR10 repository (https://github.com/anonymous-sushi-armadillo/fast_is_better_than_free_CIFAR10) as mentioned in another comment. It should only require PyTorch 1.2 and Apex, and if you really don't want to install Apex for half precision, it is very straightforward to just comment out the calls to apex and replace them with normal backwards calls (the Apex amp library is extremely unobtrusive, and so reverting to full precision as straightforward as undoing the 3 line changes described here: https://nvidia.github.io/apex/amp.html).
>
> As for your training settings, the main difference is in the learning rate schedule. Something that we noticed is that when training with piecewise learning rate for a large number of epochs (e.g. 100), there's a chance that the training process will also result in the catastrophic overfitting, resulting in the 0% PGD accuracy that you obtained (though not always, depending on the random seed). However, when this does occur, the performance notably does not degrade gracefully: the performance will go from having competitive PGD accuracy to 0% PGD accuracy within the span of a single epoch.
>
> Note that this sudden drop in pgd performance is also reflected in the training data as well, so it's easily remedied by calculating the pgd accuracy on a few training minibatches to detect overfitting and just early stopping when it's detected, if you must use a piecewise learning rate for many epochs. However this occurred very rarely for the cyclic learning rate (which is what we use and describe in the paper), and so we ended up just mentioning it as a note in the appendix, since our focus was on optimizing for the fast setting with cyclic learning rates and not the much slower setting with piecewise learning rates. In hindsight, we admit that this was probably another significant failure mode for FGSM training that past attempts have run into, and will expand upon this in a followup discussion.

---

> > ### Public Comment · ~Dinghuai_Zhang1 · 2019-09-30
> > **Probably not, after changing to cyclic I still got 0.00% robust accuracy**

---

> > > ### Public Comment · ~Tianhang_Zheng1 · 2019-09-30
> > > **I guess the key point in their training stage is *early stop***
> > >
> > > They mentioned a sudden drop in pgd performance in the training process, so they use a "early stop" test using the pgd accuracy.
> > >
> > > Extremely, if you check the pgd accuracy in every step, then the cost is almost same as (even higher than) pgd adversarial training. Then this *early stop* check becomes a little bit tricky.
> > >
> > > But it seems like the *early stop* operation *is executed every epoch in their code*. I am not sure if it can exactly find the "sudden drop" point compared with checking every step.
> > >
> > > Thanks,
> > > Tianhang

---

> > > > ### Public Comment · ~Dinghuai_Zhang1 · 2019-10-01
> > > > **I agree, but the best results in training are still much worse than baseline**
> > > >
> > > > I also notice the sudden drop. So I guess we should report the *best* results through the training process.
> > > > However, after I check the training process, I find that the best results is around
> > > >                         Clean Acc  |  PGD20 Acc
> > > > (epoch30)      73.79%       |  41.54 %
> > > > which is worse than PGD10 advtrain baseline in my codebase (about 86%, 43%). (I use res18)

---

> > > > ### Author Response · Authors · 2019-10-01
> > > > **Regarding early stopping**
> > > >
> > > > Hi Tianhang,
> > > >
> > > > For CIFAR10 models trained for 15 or 30 epochs, as reported in the paper, checking for early stopping was not necessary. We only needed to incorporate early stopping for experiments that trained for more epochs or used a larger FGSM step size. For early stopping, we only check the PGD accuracy at the end of each epoch on one minibatch, after having trained on all minibatches for that epoch. Therefore, the early stopping check adds a fairly trivial amount of computation.

---

### Public Comment · ~Chris_Finlay1 · 2019-09-29
**black box attacks (gradient-free)**

This is a very intriguing paper. It could be strengthened quite a lot by including results of gradient-free (black-box) attacks.

---

> ### Public Comment · ~Anthony_Wittmer1 · 2019-09-30
> **Concur**
>
>
> It is necessary to evaluation on the black box attacks, such as Nattack[1].
>
> When proposing a new adversarial defense, the goal is to produce a model that is robust against _all_ possible attacks within a threat model. For instance, increased robustness to FGSM attacks does not constitute progress if the model is vulnerable to PGD attacks. This is why the robust accuracy of a model is defined as the *minimum* accuracy achieved against the worst-case attack within the threat model.
>
>
> [1] NATTACK: Learning the Distributions of Adversarial Examples for an Improved Black-Box Attack on Deep Neural Networks. ICML 2019

---

### Public Comment · ~Tianhang_Zheng1 · 2019-09-30
**The implementation of PGD attack from Ln. 54 to 63 in "evaluate_cifar.py" seems to miss one clip operation in each iteration (Actually there should be two clip operations)?**

Hi everyone,

Finally, I seem to find an issue in the code:

The delta (adv perturbation) seems to be only clipped by (-epsilon, epsilon) *in each iteration* in the code (evaluate_cifar.py in the "fast_is_better_than_free_CIFAR10" directory)
However, delta also has to be clipped by *(min_pixel_value-X,  max_pixel_value - X) in each iteration* to ensure X+delta is in the range of (min_pixel_value, max_pixel_value).

This operation is done in MadryLab's code and Trades simply by clipping the X+delta not delta. Their code clips X+delta (X_adv) by (min_pixel_value, max_pixel_value) *in each iteration*, and then optimizes on the clipped X_adv in the next iteration.

**Did I make any mistake regarding this observation?** Could anyone help me double-check it?

BTW, this makes a lot of difference. For example, in one step, delta_i= 0.01, X_i = 1.0, then X_adv_i = 1.01. Without the aforementioned clip operation, the delta_i = 0.01 will be maintained. And the following step will add the perturbation on 1.01 not 1.00. Suppose the perturbation is -0.01, we will get 1.00 in the next step (no movement) instead of 0.99 (which is what we expect to see).
Thanks,
Tianhang

---

> ### Public Comment · ~Dinghuai_Zhang1 · 2019-09-30
> **Tianhang may be right...**
>
> I check the code and find some clip operations indeed miss as Tianhang said. I am not sure if this is a big problem.

---

> > ### Public Comment · ~Tianhang_Zheng1 · 2019-09-30
> > **Let's wait for the results after the authors corrected it**
> >
> > I saw the authors already corrected it in the commit history. I guess they might rerun the experiments. Lets wait for the results. I am also not very sure.

---

> ### Author Response · Authors · 2019-09-30
> **Thanks for your feedback**
>
> That's a fair point.  While we did actually have both clip operations within each iteration of PGD, we were storing the unclipped delta and clipping it before passing it through the model (so the clip between the minimum and maximum pixel range was on L55, and the epsilon clip was on L62, you can see the previous git history here: https://github.com/anonymous-sushi-armadillo/fast_is_better_than_free_CIFAR10/blob/e6032ecb1cfe32c226c9502a38f7329caafaf585/evaluate_cifar.py#L55), which is a subtle difference from other implementations. We didn't think the clipping at prediction or in delta would make much difference, especially since e.g., clipping is uncommon for CIFAR, so mainly this is an issue for MNIST.  But you're absolutely right that the procedure you suggest is indeed more correct, since otherwise there is incorrect behavior at the boundaries when delta exceeds the allowable region and only get clamped for the model prediction.  We've corrected this and updated the code in the repository.  The resulting PGD accuracies are effectively the same on CIFAR10 (reduced by 0.04%), and we'll verify our other experiments as well.  Thanks for your enthusiasm and diligence in working through this.
>
> See our response to Dinghuai for some of our thoughts on why you may be seeing different performance in your example (https://openreview.net/forum?id=BJx040EFvH&noteId=SJxjSkR0DB).  Are you able to run the code from our repository yet?

---

> > ### Public Comment · ~Tianhang_Zheng1 · 2019-09-30
> > **Thanks for your reply**
> >
> > First, thanks for your reply.
> >
> > Yep, I concur with you that this might be an issue mainly for MNIST (especially when eps=0.3, a fair number of pixels might exceed the boundaries in the optimization process). I am still not sure about the reason why we got different results on MNIST. Is it because of the different settings in the training stage or caused by this issue? I do not have the time to figure it out now, maybe I will catch it up later.
> >
> > I am glad that you will verify the experiments again (on MNIST and ImageNet). Look forward to more details.
> >
> > Anyway, this is a very interesting work.
> >
> > Thanks again,
> > Tianhang

---

> > > ### Author Response · Authors · 2019-10-03
> > > **Results are the same**
> > >
> > > Hi Tianhang,
> > >
> > > After rerunning the PGD attack with the correction in clipping, the MNIST results are largely the same (the performance for both PGD and FGSM training went down by a similar fraction of a percent).
> > >
> > > Note that the ImageNet results are unchanged, since our implementation is forked from the free adversarial training repository which clipped correctly (https://github.com/mahyarnajibi/FreeAdversarialTraining).
> > >
> > > Additionally, we've released the MNIST code as well (see https://github.com/anonymous-sushi-armadillo/fast_is_better_than_free_MNIST), where running the training script with the default parameters reproduces the results in the paper. This may help identify what caused your implementation of FGSM to fail in the past (there could be potentially other small changes which cause FGSM training to fail beyond what we ran into in the paper).

---

> > > > ### Public Comment · ~Tianhang_Zheng1 · 2019-11-04
> > > > **Thanks for the additional efforts.**
> > > >
> > > > I rechecked the code, and I could not find any mistakes in this version. So I would prefer to believe the results (are correct). Thanks for the additional results. These results are surprising and insightful, which reminds me to rethink the necessity of pgd adversarial training (from a theoretical perspective)
> > > >
> > > > Thanks,
> > > > Tianhang

---

### Public Comment · ~Xingjun_Ma1 · 2019-09-30
**Similar results have been shown in ICML2019 "On the Convergence and Robustness of Adversarial Training"**

An interesting observation!

However, we have shown similar results with a "FGSM-PGD" variant of adversarial training in our ICML2019 paper "On the Convergence and Robustness of Adversarial Training".

We show there is even a 2%-3% robustness improvement over Madry's PGD-training  (against PGD-20, epsilon=8/255) if simply use FGSM instead of PGD for the first 20 epochs (see Figure 3b).
"we replace the first 20 epochs of PGD-eps/4 training with a much weaker FGSM (1 step perturbation of size ), denoted as “FGSM-PGD”, and show its robustness and FOSC distribution in Figure 3b and 3c respectively. We find that by simply using weaker FGSM adversarial examples at the early stage, the final robustness and the convergence quality of adversarial examples found by PGD at the later stage are both significantly improved."

Basically, what we found is that it doesn't need strong adversary like PGD to solve the inner-maximization, especially in early training.  Although we didn't test further how many  (or may be "all" like in this paper) epochs one can use FGSM to get the best robustness, the main finding in this paper is somewhat similar to ours. In the paper, we have already theoretically proved that the convergence of min-max adversarial training only requires the inner maximization is solved up to a certain precision  (by PGD or FGSM).

It would be interesting to see some empirical results and discussion on this. For example, what would happen if you replace the last 10 epochs of training by PGD.

In the meantime, I would like to clarify that this does not lower the contributions of this paper, as we haven't considered to replace PGD COMPLETELY with FGSM.

Link to our paper: http://proceedings.mlr.press/v97/wang19i/wang19i.pdf

Thanks.

---

> ### Public Comment · ~Tianhang_Zheng1 · 2019-09-30
> **Interesting work**
>
> Interesting and solid work!
>
> A small question: what is the reason for the 2%-3% improvement?
>
> Thanks,
> Tianhang

---

> > ### Public Comment · ~Xingjun_Ma1 · 2019-10-05
> > **Diversity in early training helps robustness generalization**
> >
> > Hi Tianhang,
> > We find, in early training when the network still struggles to converge, diversity (of adversarial training examples) is crucial for robustness generalization. However, at a later stage, strong advs examples that are of high convergence (inner max) quality become necessary. The random start normally does the diversity trick, but can still be improved by a weak adversary such as (eg. FGSM).
> >
> > Thanks,
> > Xingjun

---

### Public Comment · ~Anthony_Wittmer1 · 2019-10-07
**Why does the early-stop strategy help so much?**

From the public comments, it seems that early-stop strategy helps the robustness so much.

Why does the missing of the early-stop strategy  make a great difference for the robustness? Do the authors have any good insights?

---

### Author Response · Authors · 2019-11-11
**Updated paper**

In light of the discussion with the public and the feedback from the reviewers, we have uploaded a revised version of the paper. The main changes can be summarized as follows:

+ We've added the discussion and comparison to R+FGSM from Tramer et. al (Appendix A)
+ We've added the experiments showing the full effect of step size across a wide range on CIFAR10 (Appendix C)
+ We've included the discussion and experiments which attempt to combine the DAWNBench optimization tricks with free adversarial training on ImageNet (Appendix E)
+ We've added training time to Table 4 as requested
+ The main text of the paper has been adjusted to reflect these additions

If there are any unsettled concerns or comments about the paper, we eagerly await further discussion from the reviewers.

---

### Public Comment · ~Sandesh_Kamath1 · 2020-04-23
**Related work citation request**

Dear Authors,

Our ICML 2019 workshop paper “Better Generalization with Adaptive Adversarial Training” proposes a frugal adversarial training with FGSM perturbations that achieves PGD-robustness comparable to your method and gives better generalization simultaneously.

https://sites.google.com/view/icml2019-generalization/accepted-papers
https://drive.google.com/file/d/1TtuScpA9luvOrAoQTOYjSVgyuPkt7XyV/view?usp=sharing

We would greatly appreciate it if you could cite our work and let us know your comments.

Thanks,
Sandesh Kamath

---

### Decision · Program_Chairs · 2019-12-19

**Decision:**

Accept (Poster)

**Comment:**

This paper provides a surprising result: that randomization and FGSM can produce robust models faster than previous methods given the right mix of cyclic learning rate, mixed precision, etc. This paper produced a fair bit of controversy among both the community and the reviewers to the point where there were suggestions of bugs, evaluation problems, and other issues leading to the results. In the end, the authors released the code (and made significant updates to the paper based on all the feedback). Multiple reviewers checked the code and were happy. There was an extensive author response, and all the reviewers indicated that their primary concerns were address, save concerns about the sensitivity of step-size and the impact of early stopping.

Overall, the paper is well written and clear. The proposed approach is simple and well explained. The result is certainly interesting, and this paper will continue to generate fruitful debate. There are still things to address to improve the paper, listed above. I strongly encourage the authors to continue to improve the work and make a more concerted effort to carefully discuss the impacts of early stopping.